# Study on the Damage Mechanism of TiN/Ti Coatings Based on Multi-Directional Impact

**Zhihao Fang [1], Jiao Chen [2], Weifeng He [1],*, Zhufang Yang [1], Zhanwei Yuan [3], Mingrui Geng [1] and Guangyu He [1],***

[1]  Science and Technology on Plasma Dynamics Laboratory, Air Force Engineering University, Xi'an 710038, China; fanghugo@163.com (Z.F.); yangzf1113@126.com (Z.Y.); gmr_halak8@163.com (M.G.)
[2]  State Key Laboratory for Manufacturing Systems Engineering, Xi'an Jiaotong University, Xi'an 710049, China; chenqiao1111@stu.xjtu.edu.cn
[3]  School of materials Science and Engineering, Chang'an University, Xi'an 710061, China; yuanyekingfly@163.com
*   Correspondence: hehe_coco@163.com (W.H.); hegy_22@126.com (G.H.); Tel.: +86-29-84787527 (G.H.)

**Abstract:** TiN/Ti coatings have great application potential in improving aero-engine server lives in a dusty environment. However, the damage behavior and mechanism of the coating and substrate under high impact speed and multi-direction loading conditions has scarcely been investigated. In this paper, TiN/Ti coatings were deposited on Ti6Al4V alloys by a magnetic filter cathode vacuum arc. Multi-directional impact tests were carried out by a gas gun system at, 45°, 60°, and 90° with a velocity of 330 m/s. The damage behaviors and mechanisms of the TiN/Ti coatings were investigated and revealed by researching the damage morphology, crack propagation, and stress distribution. The results show that plastic deformation occurs both in the coatings and the substrates under high speed impacting. Cracks extend vertically downward in the TiN layer first and are deflected at the Ti layer when the driving force is not enough. Circular cracks and radical cracks are found to form network cracks on the surface of the coating and the shear stress loaded by the particles, which drives cracks' propagation is the main reason for the peeling off on the coatings.

**Keywords:** TiN/Ti coatings; multi-directional impact test; high erode speed; damage mechanism

## 1. Introduction

Solid particle erosion (SPE) is very common in industry. Especially for helicopters and transporters serving in the desert, the problem of solid particle erosion is more prominent and urgent. An anti-erosion coating has been considered as an effective way to improve the server life of aero-engines in the past decades [1–4]. Coatings also face erosion problems brought by solid particles, and the knowledge of the performance of the coating during erosion remains insufficient due to the complexity of the damage mode and mechanism [5,6]. Thus, studies on the coating erosion damage mechanism have always been the focus in the past decades.

The mechanism of solid particle erosion (SPE) is mainly composed of two parts: The ductile and brittle mechanisms of SPE [5,7–10]. Ductile SPE mechanisms are characterized by material removal through plowing or cutting by the impacting particles, and erosion rates reach a maximum at low angles of incidence [11–13]. Early models of brittle SPE mechanisms were mostly based on the intersection and branch of conical Hertzian cracking [14,15]. With the development of electron microscopy techniques, erosion mechanisms of brittle SPE by elasto-plastic indentation were developed later [16,17]. Sand erosion is widely used to evaluate coatings' durability [7,18]. However, the particle is massive and discrete and the velocity and impact angle of the particle cannot be precisely

controlled [19,20]. This leads to only the macro erosion results being obtained. The single particle impact test as a precise measurement can overcome those shortcomings. The damage mechanisms of anti-erosion coatings in single particle impact tests have been fully studied in the literature [8,21–23]. It is believed that anti-erosion hard coatings mainly fail in brittle fracture. However, elasto-plastic deformation also occurs on the surface of coatings during impact. The intersection and branch of the cracks in the coatings are the main reason for the material removal. However, research has mostly focused on medium and low impacting speeds and the vertical impacting angle. However, in actual service conditions, particles can reach very high speeds [24], and the impact angle is also multi-directional. The coating performance under these service conditions is quite different. Thus, the damage behavior and failure mechanism of multilayer coatings under multi-directional and high impact velocity still need to be further explored.

In this article, a TiN/Ti multilayer coating was deposited on the surface of Ti6Al4V alloy by the physical vapor deposition (PVD) method. The mechanical properties of the coating were characterized by nano-indentation and the scratcher. Coatings were eroded by 1-mm diameter steel spheres at a velocity of 330 m/s and at impact angles of 45°, 60°, and 90°. The damage morphology of the coating surface was observed by scanning electron microscopy to investigate the damage behavior of the coating. The cross section of the coating was obtained by a focused ion beam experiment to investigate the crack network under the surface. The stress distribution of the impact area was investigated by finite element simulation to further explain the failure behavior of the TiN/Ti multilayer coating. The results of this paper are intended to provide references for understanding the impact damage behavior of multilayer erosion resistant coatings under high-speed and multi-directional impact.

## 2. Materials and Methods

### 2.1. Sample Preparation

The TiN/Ti coatings were deposited on the surface of the Ti6Al4V substrates. The composition of the Ti6Al4V alloy is illustrated in Table 1. Ti6Al4V alloy materials were manufactured to a size of 50 mm × 20 mm × 3 mm. Samples were polished to achieve a surface roughness ($R_a$) under 0.02 μm. Then, those substrates were cleaned by an ultrasonic cleaner with acetone for 15 min and dried with alcohol before deposition.

**Table 1.** Chemical composition of the Ti6Al4V titanium alloy.

| Element | Ti | Al | V | Fe | C | O | N | H |
|---|---|---|---|---|---|---|---|---|
| **Content (wt.%)** | Bal. | 5.70 | 4.00 | 0.10 | 0.02 | 0.05 | <0.01 | <0.001 |

The microstructure and mechanical properties of the coating are affected by the deposition temperature, bias pressure, arc current, and nitrogen partial pressure. With an increase of the bias pressure, the temperature of the substrate increases, which leads to an increase of the density of the film. However, when the bias pressure is too high, the substrate will soften and anneal due to overheating, and the interface stress of the film and substrate will increase with its hardness and adhesion, displaying a poor performance. The arc current was maintained to melt and evaporate target. It was found that the deposition rate increases with the increasing of the arc current. However, when the arc current is too large, it is easier to produce a droplet, which leads to a decrease of the mechanical property of the coating. Before the deposition, in the multi-arc ion plating system, Ar gas with a flow of 80 sccm was introduced into the chamber with a bias voltage of −500 V and temperature of 200 °C to clean the substrates for 30 min and then ion implantation was carried out with the vacuum degree of $1.0 \times 10^{-3}$ Pa, voltage of 10 KV, and ion beam of 6 mA. Then, in the magnetic filter cathode vacuum arc system, the Ti layer was deposited on the substrate as the transition layer. Pure Ti targets (99.99%) and $N_2$ gas (99.99%) were introduced into the chamber to obtain the TiN layer. As is shown in Table 2, the TiN layer was deposited with an $N_2$ pressure of 1.0 Pa, negative bias voltage of −200 V, and

arc current of 100 A. In addition, Ti sputtering was performed twice during each TiN layer deposition with an arc current of 110 A. Negative bias voltages of −800, −600, and −400 V were conducted in sequence for 2 min, respectively, during Ti sputtering. The total thickness of the TiN/Ti coating was 16 μm with a thickness ratio of 9:1 and period numbers of 2.

**Table 2.** Deposition parameters of TiN layer.

| Deposition Parameters | $N_2$ Pressure (Pa) | Bias Voltage (V) | Arc Current (A) | Coating Thickness (μm) |
|---|---|---|---|---|
| value | 1.0 | −200 | 100 | 16 |

### 2.2. Mechanical Property Testing

Scanning electron microscopy (SEM) (MIRA 3, TESCAN, Brno, Czech Republic) was carried out to observe the surface and cross-sectional micrographs of the TiN/Ti coatings. The phase compositions of the TiN/Ti coatings were measured by X ray-diffraction (X·pert, Philips, Amsterdam, Netherlands) with a Cu target K alpha peak, 40 KV working voltage, and 40 mA working current. The scanning step length (2θ) was 0.02° and the scope of the diffraction angle was 20° to 90°.

The adhesion of the TiN/Ti coatings was measured by using the WS-2005 scratch tester (Zhongke Kaihua Technology Development, Lanzhou, China). The cohesion adhesion test was carried out with a loading rate of 100 N/min, load of 100 N, and scratch length of 5 mm. The microscopic morphology of the scratch was observed by SEM.

The Young's modulus and hardness of the TiN/Ti coatings were measured by a nano-indenter (Agilent U9820A, Agilent, Santa Clara, CA, USA) with Berkovich indenter (Agilent, Santa Clara, CA, USA). Five indentations were performed with a continuous load mode to obtain the Young's modulus and hardness of the TiN/Ti coatings. During the test, to reduce the effect of the substrate, the maximum indentation depth was no more 1000 nm, which was less than 1/10 of the thickness of the coating. The displacement–load curve was obtained by measuring the load and the depth of the compression. The hardness and Young's modulus of the TiN/Ti coatings were calculated by Oliver-Pharr's method [25] according to the displacement–load curve.

### 2.3. Multi-Directional Impact Testing

The gas gun experimental system was used to accelerate steel ball particles and impact the TiN/Ti coatings at different impact angles, as shown in the Figure 1. The gas gun experimental system was mainly composed of four parts, namely the gas source, control system, speed measuring device, and sample fixture. The impact particles were GCR15 spherical particles with a diameter of 1 nm, and their composition parameters are shown in Table 3. A particle was loaded into the projectile tray, which was put into the track by unscrewing the sealing sleeves and accelerated by the expansion of compressed nitrogen. The impact angle was changed by adjusting the deflection of the fixture. To investigate the high-speed damage of the coating at different impact angles, impact tests were conducted at a 330 m/s incidence speed and incident angles of 45°, 60°, and 90°. Three impact tests were carried out under each incident angle.

**Table 3.** Chemical composition of the GCR15 steel ball.

| Element | Fe | C | Cr | Si | Mn | Mo | Ni | Cu |
|---|---|---|---|---|---|---|---|---|
| Content (wt.%) | Bal. | 1 | 1.65 | <0.35 | <1 | <0.1 | <0.3 | <0.25 |

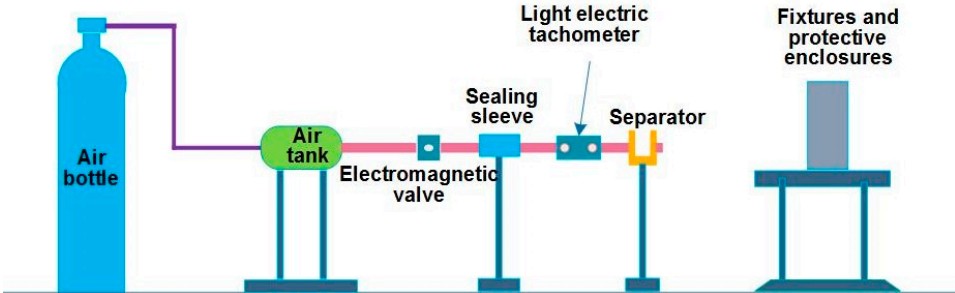

**Figure 1.** Diagram of the gas gun impact test system.

Scanning electron microscopy (SEM), back-scattered electron imaging (BSE), and energy dispersive spectrometry (EDS) were used to analyze the microstructure and the typical elements' distribution on the surface of the coating.

## 3. Results and Discussion

### 3.1. Structure and Phase Analysis

As shown in Figure 2, the surface of the TiN/Ti coatings was golden and it can be observed that the coatings are composed of the TiN layer and Ti layer with two cycles. TiN layers and Ti layers were deposited alternately on the substrate with a modulation ratio of 9:1 and the total thickness of the coating was 16 μm.

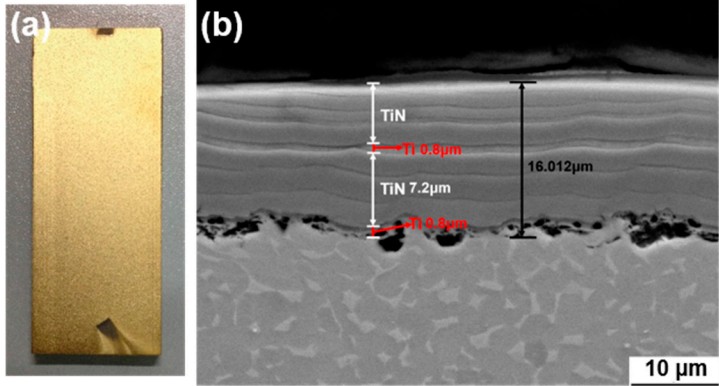

**Figure 2.** The microtopography of the surface (**a**) and cross-section (**b**).

The X-ray powder diffraction (XRD) patterns of TiN/Ti multilayer coatings are shown in Figure 3. The diffraction peaks are in accordance with the cubic TiN after checking with the JADE software (version 6.5). The (111), (200), (220), (311) diffraction peaks were identified from the TiN/Ti coatings. The strongest diffraction peak corresponds to the (111) plane of the TiN. During the deposition, the preferred orientation of the TiN coatings was determined by the competition between the surface energy and the strain energy. It was found that the (200) plane has the lowest surface energy while the (111) plane has the lowest strain energy. In the early stage of the deposition, the strain energy was small, and the surface energy was dominant. The coating was deposited in the (200) preferred orientation. Besides, researches have shown that strain energy gradually increases with the increasing of the thickness. Thus, as the deposition occurs, the strain energy is large and dominant, in which case the (111) preferred orientation is predicted [26–28]. This leads to the (111) plane having the highest peak and the (200) plane having a lower diffraction peak. The appearance of other lower peaks may be due to the incomplete orientation of the initial grain growth process. The preferred orientation of the

TiN was determined by calculating the texture coefficient. The value of the texture coefficient $T(hkl)$ was calculated using Equation (1):

$$T(hkl) = \frac{I(hkl)/I_0(hkl)}{\frac{1}{n}\sum I(hkl)/I_0(hkl)} \tag{1}$$

where $I(hkl)$ is the integral intensity of the coating on the $(hkl)$ reflector. $I_0(hkl)$ is the integral intensity of the standard powder on the $(hkl)$ reflector. $n$ is the number of diffraction peaks. The texture coefficient value of the (111) plane of this coating in this paper is 2.3, larger than 1, which means that the (111) plane is the privileged orientation of TiN.

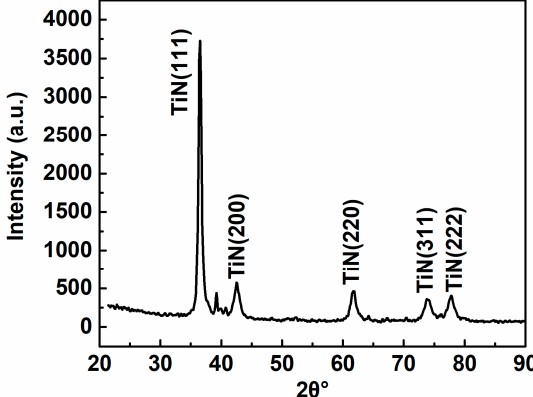

**Figure 3.** XRD diffraction pattern of the TiN/Ti multilayer coating.

## 3.2. Mechanical Properties

The scratch method, as a widely accepted adhesion force test method, was applied in this work. The acoustic emission spectra, friction force curve, and scratch track morphology were obtained in the scratch test. To investigate the adhesion, five scratch tests were performed for each sample. As shown in Figure 4, it was found that the width of the scratch track increases with the load increasing. When the load reaches 53 N, there is a sudden change in the acoustic emission signal and the friction force curve, while the scratch track is still very smooth, and the coating still has good integrity. When the load reaches 70 N, the friction force curve and acoustic emission signal fluctuate fiercely, and the coating begins to peel off from the surface. In this case, the load applied reaches the critical load of the coatings. The average values of the adhesion results are shown in Table 4. The adhesion strength of the coating is 72 N. Since the surface of the substrate was subjected to Ar ion irradiation and ion implantation during the pretreatment, the adhesion of the coating reaches quite a high level.

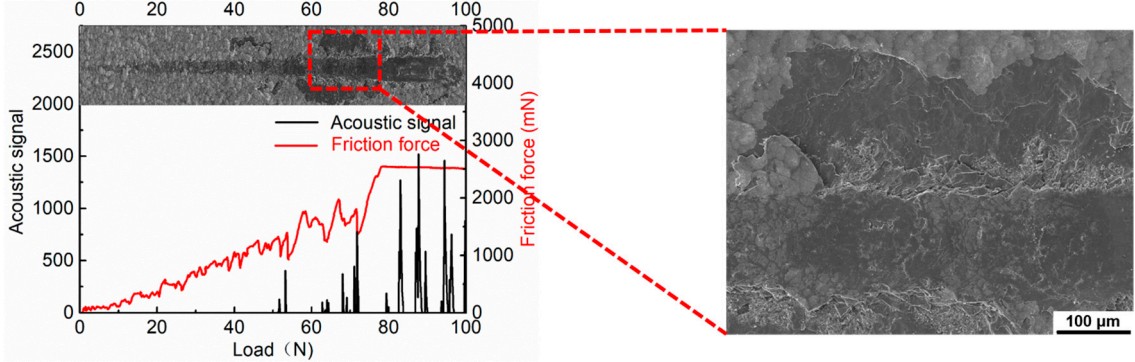

**Figure 4.** Scratch morphologies, acoustic emission spectra, and friction force curve and SEM image.

**Table 4.** Mechanical test results of the coating.

| Adhesion (N) | Hardness (GPa) | Young's Modulus (GPa) |
| --- | --- | --- |
| 72. 5 ± 2.5 | 27.06 ± 0.59 | 365.62 ± 4.94 |

In the nano-indentation test, the fixed load mode was applied, and the fixed load used was 100 mN. As shown in Figure 5, the indentation depth was 450 nm, which is less than the 1/10 of the thickness of the coatings to reduce the influence of the substrate. The hardness and Young's modulus of the coating were calculated according to the displacement–load curve. The nano-indentation test was performed five times, and the average values and corresponding standard deviations of the hardness and Young's modulus are presented in Table 3. The hardness was about 27 GPa and the Young's modulus was about 365 GPa.

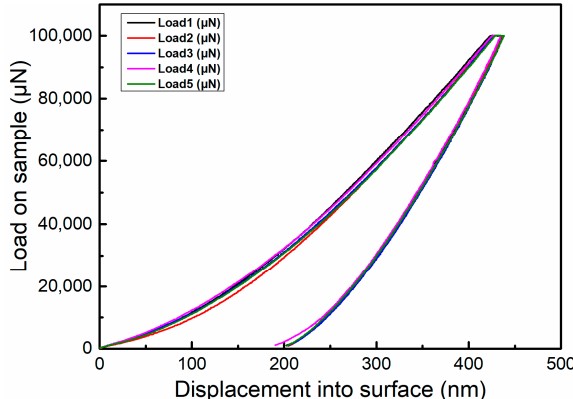

**Figure 5.** Nano indentation displacement–loading curves of the TiN/Ti coatings.

## 3.3. Damage Morphology

In order to investigate the damage mechanism of the coating at high impact speed under different angles of attack, the damage morphology was observed by scanning electron microscopy. A series of images illustrating the damage morphology of the TiN/Ti coating impacted at different impact angles with an eroding speed of 330 m/s are presented in Figure 6. It is easy to see that deformation occurred on the surface of the coatings, which led to the formation of impact pits. Circular cracks were found distributed around the contact point. The direction of particle motion is the positive half axis. The circular crack density in the positive half axis is lower than the opposite side in damage morphology at 45° and 60°. However, the crack density is relatively uniform in the damage morphology at the 90° impact angle. In addition, radial cracks were also found distributed on the surface of the coatings. The circular cracks and the radial cracks form network cracks as shown in Figure 6. There are large peeling areas at the distance of the contact radius to the contact center, which expands in the direction of the particle's motion. The peeling area is the largest at 45°, followed by 60°, and is the smallest at 90°.

In order to investigate the crack propagation in the coating interface, focused ion beam (FIB) experiments were conducted. Figure 7a shows the crack propagation in the coating interface at the 45° impact angle. It is obvious that the cracks first extend vertically downward from the surface of the top TiN layer. They deflect when they reach the Ti layer and begin to grow along the interface. Similar to the phenomenon at 45°, cracks at the 60° (Figure 7b) impact angle first propagate vertically downward in the TiN layer and then deflect when they reach the interface of the TiN layer and Ti layer. However, the length of the cracks deflecting along the Ti layer at 60° is shorter than 45°. At the 90° (Figure 7c) impact angle, most cracks extend vertically downwards and almost do not deflect in the Ti layer. To sum up, the distance of the cracks deflecting in the interface along the Ti layer reduces as the impact angle increases. The coating peels off in layers at the broken surface, which may lead to coalescence of the interface cracks and circular cracks.

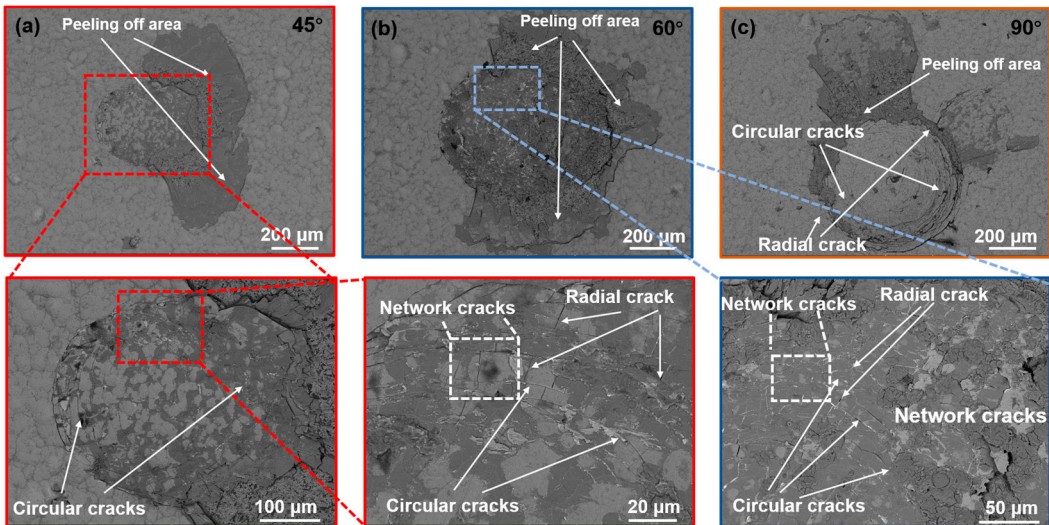

**Figure 6.** SEM micrographs of TiN/Ti coatings impacted under different angles: (**a**) Impacted under 45°, (**b**) impacted under 60°, and (**c**) impacted under 90°.

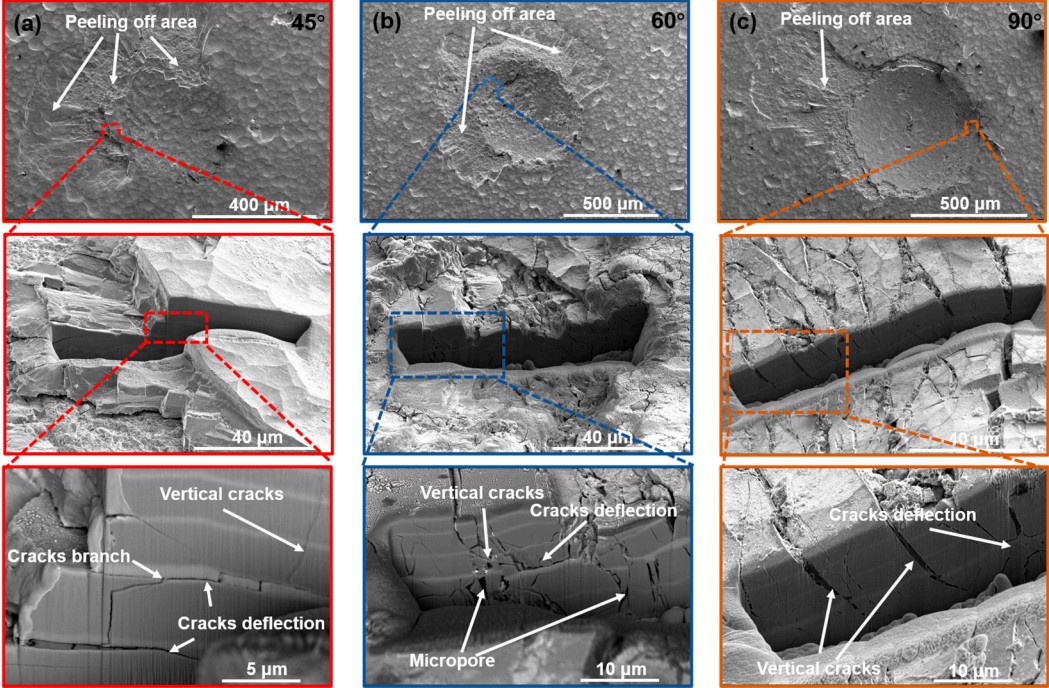

**Figure 7.** Impact damage morphologies and the sectional images of eroded of TiN/Ti coatings impacted under different angles: (**a**) Impacted under 45°, (**b**) impacted under 60°, and (**c**) impacted under 90°.

### 3.4. Coating Impact Damage Mechanism

Finite element modelling was used to investigate the stress fields and dynamic responses of TiN/Ti coatings subjected to particle multi-direction impacting with high speed. The eroding particle was modeled as a ball of GCR15 with a diameter of 1 mm. The coating was modeled as an alternating superposed structure consisting of TiN layers and a Ti layer. The period number of the coating was 2 with a total thickness of 18 μm. In this modelling, four types materials were implemented. The Ti6Al4V substrate, Ti layers, and the GCR15 steel ball were modelled as an elastic-plastic material. The TiN layers were defined as a perfectly elastic material. The detailed material parameters are presented in Table 5. To investigate the high-speed impact damage of coatings at different impact angles, the initial

velocity of the eroding particle was 330 m/s, and the impact angle was 45°, 60°, and 90°. The coating and substrate were meshed with a four-node axisymmetric minus integral quadrangle element and the ball was meshed with a triangular axisymmetric element. The mesh of the contact surface on the coating was refined to improve the computational accuracy.

**Table 5.** Material parameters of the Finite element modelling.

| Material | Density (kg/m³) | Young's Modulus (GPa) | Poisson's Ratio | A (MPa) | B (MPa) | n | C |
|---|---|---|---|---|---|---|---|
| Ti6Al4V | 4428 | 113.8 | 0.34 | 2415 | 2402 | 0.93 | 0.014 |
| Ti | 4500 | 112.0 | 0.34 | 1647 | 1638 | 0.16 | 0.014 |
| GCR15 | 7800 | 208.0 | 0.30 | 1000 | 1100 | 0.19 | 0.027 |
| TiN | 5400 | 400.0 | 0.27 | – | – | – | – |

*A*: yield strength in indoor temperature, *B*: strain rate hardening index, *n*: strain hardening, *C*: heat softening effect.

The displacement of the eroded center is displayed in Figure 8. It can be observed that the displacement at 45° has the smallest amplitude and that at 90° it is the largest. Figure 8 clearly shows that the plastic deformation on the surface of the coating is mainly determined by the perpendicular incident component of the particle. Stress S11 (the stress along *x* direction) on the surface of the coating impacted under 45°, 60°, and 90° at 20 ns is shown in Figure 9. As presented in Figure 9, there are compressive stresses near the impact center and high tensile stress peeks distributed at the distance outside from the impact center of about a contact radius, which is consistent with findings in other studies [29–31]. Those areas of high tensile stresses S11 are the main reason for the development of surface cracks. This type of stress pattern leads to the formation of circular cracks, which has been reported in many researches [2,8,30]. It can clearly be seen that the amplitudes of the tensile stresses at 90° are larger than that of 60° and 45°. The direction of the particle motion is taken as the positive half axis. The amplitude of the tensile stresses in the negative half axis of the contact point is larger than the opposite side at the 45° and 60° impact angles. However, at the 90° impact angle, the amplitude of the tensile stress on both sides is equal. This stress distribution on the coating surface may account for the distribution of the circular cracks on the surface, as shown in Figure 7. It is not difficult to find that the density of circular cracks formed in the region with higher tensile stress is relatively higher than that in the lower tensile stress region, as shown in Figure 6a,b. When the surface of the coating deforms, the dislocation of the crystal can be blocked at the grain boundary, which leads to the stress concentration at the grain boundary [32]. The stress at the grain boundary in the higher tensile stress region is easier to reach the grain boundary strength. Therefore, the crack density in the higher tensile stress region is larger than that in the lower tensile stress region.

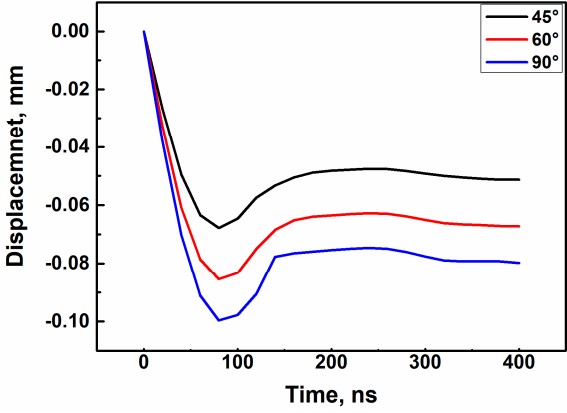

**Figure 8.** Displacement of the TiN/Ti coatings during impact.

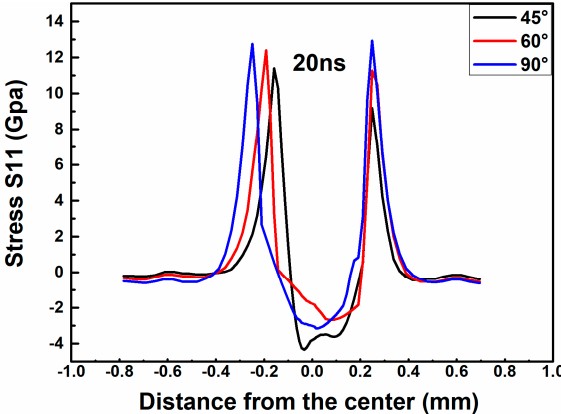

**Figure 9.** Stress fields of the TiN/Ti coating under different impact angles at 20 ns.

The microstructure of the TiN layer is columnar, and its growth direction is perpendicular to the substrate surface [33]. Initial cracks come into being at the grain boundary first, and then crackle spreads along grain boundaries occur for the tensile stress on the surface of the TiN/Ti coatings. Therefore, most cracks grow normal to the interface in the TiN layer first, as shown in Figure 7. The Ti layer as a typical ductile material can absorb the energy of the crack propagation. When the driving force for crack growth is not high enough, crack deflection and branching occurs at the interface of the Ti layer. However, some cracks grow through the Ti layer due to the high tensile stress, which provides enough driving force for crack growth. For this reason, cracks near the impact center at the 90° impact angle are easier to grow through the Ti layer with higher tensile stress. While, crack deflection and branching are more likely to occur at 45° and 60° where the tensile stress is lower. Incoordination in the deformation process of the bottom TiN layer and substrate is the main reason why internal cracks and micropores occur in the bottom TiN layers. During the impacting, plastic deformation occurs both in the coating and the substrate. The deformation performance of the two materials is quite different. The stress concentration phenomena easily occurs at the border between the bottom TiN layer and the substrate. Therefore, there is usually high tensile stress in the subsurface of the TiN layer, which has been mentioned in many reports [2,8]. When the stress concentration reaches the material strength of the TiN, micropore and microcracks form in the bottom TiN layer and are enlarged further, driven by the tensile stress (Figure 7b) [34]. Figure 10 shows the dynamic process during the impacting. It can be found that this process is mainly composed of incident, rebound, and separation. There is also relative sliding friction between the particles and coatings during incident and rebound. This relative sliding friction loads shear stress on the coating surface. The shear stress appears to favor the growth of cracks on the coatings, and crack propagation and intersection are responsible for the peeling off on the coating in the network cracks' area. The distance of relative sliding at 45° is the largest, followed by 60°, and no relative sliding occurred at 90°, which also corresponds to size of the peeling-off area under corresponding impact angles.

To sum up, the Ti layer plays an important role in the prevention of crack propagation normal to the substrates. Crack deflection and branching occurs at the Ti layer when the tensile stress is not large enough; propagation and intersection are the main reason for the material removal of coatings.

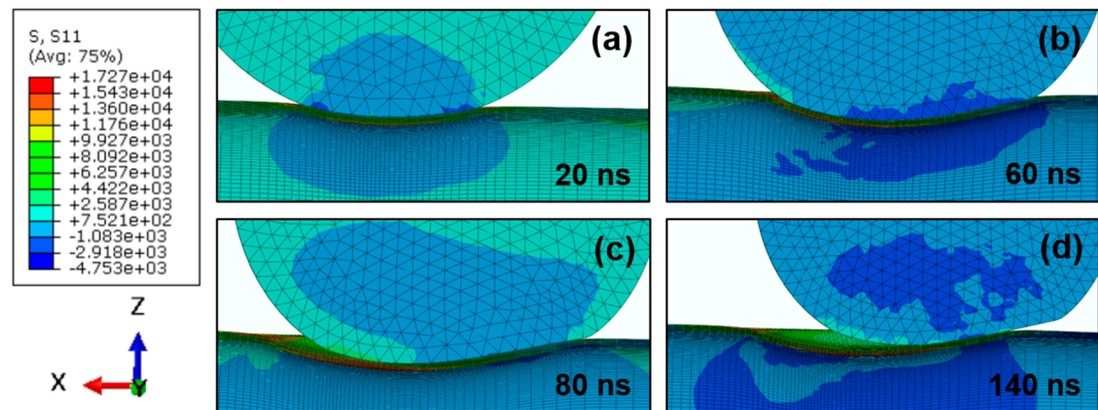

**Figure 10.** Displacement and stress fields of the TiN/Ti coating in four periods: (**a**) At 20 ns (incident time), (**b**) at 60 ns, (**c**) at 80 ns (deepest time), and (**d**) at 140 ns (separated time).

## 4. Conclusions

TiN/Ti coatings were deposited on Ti6Al4V substrates by the PVD method. Impact tests under different angles were performed. The damage morphologies were analyzed to investigate the damage mechanism of the coating. The results can be concluded as follows:

- Plastic deformation occurs both in the coatings and substrates and increases with an increase of the impact angle during high-speed impact.
- The high tensile stress on the coating surface is responsible for the circular crack formation, and the crack density increases with the increasing of the tensile stress.
- Cracks deflect at the Ti layer because the Ti layer has good toughness and can absorb a certain amount of fracture work. Crack deflection and branching more easily occurs when the driving force is lower (at 45° and 60°), but cracks grow through the Ti layer when the driving force is enough.
- The relative sliding between the particle and coating on the network area is the main reason for the peeling off on the coatings.

**Author Contributions:** Conceptualization, G.H. and Z.Y. (Zhufang Yang); Writing—Original Draft Preparation, Z.F.; Software, Z.Y. (Zhanwei Yuan); Formal Analysis, Z.F.; Writing—Review and Editing, J.C.; Investigation, M.G.; Project Administration, W.H.

**Funding:** This research was funded by the National Natural Science Foundation of China (No. 51405506) and National Science and Technology Major Project (2017-VII-0012-0107).

**Acknowledgments:** Thanks State Key Laboratory for Manufacturing Systems Engineering for helps in morphological characterization.

**Conflicts of Interest:** The authors declare no conflict of interest.

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
