# Peer review of "Study on the Damage Mechanism of TiN/Ti Coatings Based on Multi-Directional Impact"

_coatings, doi:10.3390/coatings9110765_

Round 1
Reviewer 1 Report
Before publication of the manuscript, authors should be considered the following comments:
- Both in the description of the text and in Figure 1, the authors present the air-gun impact test system, while in the conditions of the experiment they write the use of compressed nitrogen. (line 109)
- In my opinion, authors misinterpret that privileged orientation of TiN in plane (111) is based on high peak intensity of Ti (111). To confirm this thesis, authors should describe a specific crystallographic relationship. In the discussion the authors have not refer to the reason the low intensity of the other TiN peaks and their really big half-width. It is an important element affecting the performance of multilayer coatings with different chemical and phase composition produced under specific PVD process conditions. Authors also does not describe how specific parameters of the PVD process affect the performance of the coatings (layers) produced, providing unconditionally only the values of the process parameters used. On Fig. 3 we can see a very high porosity on the coating / substrate conection area and a very nice structure of the multilayer coating, with a very large thickness in relation to the PVD process. This porosity disqualifies these coatings for industrial applications. With well parameters of the PVD process, this porosity could be minimized, and before testing the other mechanical and erosion properties of the produced coatings, the authors should focus on eliminating this porosity. As we can see, the use of Ti interlayer alone is not enough. What is the point of examining the functional properties of coatings in their volume, since the coating is not consistent with the substrate? Scratch test shows the effect on displacement into surface multilayer coatings to a depth of 400 nm when maximum load is reached (Fig. 5). It is result of the hard TiN layers participation in the structure of the thick coating, which does not mean that this coating has high adhesion (which can see in Fig. 2).
- What does stress S11 mean on the y axis in Fig. 9 ? There is no description in the content of the manuscript.
- Hypotheses about the generation of a specific value of tensile residua stress are purely hypothetical without conducting research results.
- The second conclusion is hypothetical not supported by the own research results.
- In Fig. 4 incorrectly it was specified the friction force unit (g)
- Table 1 and table 3 (chemical composition). What % (at. or wt.) ?
- Many mistakes of Mpa there are in manuscript and Table 5. It should be MPa.
- Why the authors write that TiN/Ti coatings are deposited on Ti6Al4V alloys (as substrates in plural). In these case, how many Ti6Al4V alloys were there?
Reviewer 2 Report
Review:
- Page 2, line 48 – start sentence with capital letter („fracture. While“)
- Page 2, line 67 - use capital letter for title of chapter („Sample preparation“)
- Page 2, line 70 – use subscript „a“ in Ra
- table 1 and table 3 – explain „Content“ (mass, volume or other)
- Use capital letter for word Young’s modulus (line 95, 97, 150, 152, …)
- Page 5, line 153 - unit is GPa (not Gpa)
- Figure 4 – unit for friction force is N (not g)
- Page 6, line 169 - Figure 6 (not Figure 8)
- Table 5 - unit for density kg/m3 (not kg/m3); GPa (not Gpa), MPa (not Mpa)
Round 2
Reviewer 1 Report
Congratulation. Accept.